# Multidisciplinary Approach to the Diagnosis and Therapy of Mycosis Fungoides

**DOI:** 10.3390/healthcare11040614

**Published:** 2023-02-18

**Authors:** Paola Vitiello, Caterina Sagnelli, Andrea Ronchi, Renato Franco, Stefano Caccavale, Maria Mottola, Francesco Pastore, Giuseppe Argenziano, Massimiliano Creta, Armando Calogero, Alfonso Fiorelli, Beniamino Casale, Antonello Sica

**Affiliations:** 1Dermatology Unit, University of Campania Luigi Vanvitelli, 80131 Naples, Italy; 2Department of Mental Health and Public Medicine, University of Campania Luigi Vanvitelli, 80131 Naples, Italy; 3Pathology Unit, Department of Mental and Physical Health and Preventive Medicine, University of Campania Luigi Vanvitelli, 80131 Naples, Italy; 4Department of Heart Surgery and Transplantations, AORN Dei Colli-V Monaldi, 80131 Naples, Italy; 5Radiotherapy Unit, Emicenter, 80020 Naples, Italy; 6Department of Neurosciences, Reproductive Sciences and Odontostomatology, University of Naples Federico II, 80131 Naples, Italy; 7Department of Advanced Biomedical Sciences, University of Naples Federico II, 80131 Naples, Italy; 8Thoracic Surgery Unit, University of Campania Luigi Vanvitelli, 80131 Naples, Italy; 9Department of Pneumology and Tisiology, AO Dei Colli-V. Monaldi, 80131 Naples, Italy; 10Department of Precision Medicine, University of Campania Luigi Vanvitelli, 80131 Naples, Italy

**Keywords:** CTCL1, primary cutaneous T-cell lymphoma, mycosis fungoides, multidisciplinary team approach, early mycosis fungoides

## Abstract

Mycosis fungoides is the most common primary cutaneous T-cell lymphoma, characterized by skin-homing CD4+ T cells derivation, indolent course, and low-grade of malignancy. Mycosis fungoides’s classic type typically onsets with cutaneous erythematous patches, plaque, and tumor. In WHO-EORTC classification, folliculotropic mycosis fungoides, pagetoid reticulosis, and granulomatous slack skin are recognized as distinct variants of mycosis fungoides, because of their clinical and histological features, behavior, and /or prognosis. Mycosis fungoides often shows diagnostic difficulties, due to its absence of specific features and lesional polymorphism. A patient’s treatment requires staging. In about 10% of cases, mycosis fungoides can progress to lymph nodes and internal organs. Prognosis is poor at advanced stage and management needs a multidisciplinary team approach. Advanced stage disease including tumors, erythroderma, and nodal, visceral, or blood involvement needs skin directed therapy associated with systemic drugs. Skin directed therapy includes steroids, nitrogen mustard, bexarotene gel, phototherapy UVB, and photochemiotherapy, i.e., total skin electron radiotherapy. Systemic therapies include retinoids, bexarotene, interferon, histone deacetylase inhibitors, photopheresis, targeted immunotherapy, and cytotoxic chemotherapy. Complexity of mycosis fungoides associated with long-term chronic evolution and multiple therapy based on disease stage need a multidisciplinary team approach to be treated.

## 1. Introduction

Mycosis fungoides (MF) is the most common primary cutaneous T-cell lymphoma (CTCL), accounting for almost 50% of all CTCLs, with an annual incidence of 0.3–0.5 new cases per 100,000 inhabitants. There is a male predominance with a median age between 55 and 60 years, but children and adolescents may also be affected. MF is a low-grade lymphoma originating from skin homing CD4+ T cells. The etiology remains largely unknown, and the identification of genetic mutations and environmental factors requires further research.

Clinically, MF typically presents with cutaneous erythematous patches, plaques, and tumors. Nevertheless, atypical variants of MF exist, characterized by different clinical behavior, prognosis, and histological features. The World Health Organization (WHO)/European Organization for Research and Treatment of Cancer (EORTC) defines three subtypes of MF: folliculotropic MF, pagetoid reticulosis, and granulomatous slack skin [1].

The clinical findings of MF may be extremely heterogeneous, and several different clinical morphologies have been described, including bullous, hypopigmented, ichthyosiform, palmo-plantaris (keratoderma-like), pigmented purpuric dermatosis-like, papular, poikilodermatous, psoriasiform, pustular, and verrucoid.

The diagnosis of MF is challenging, as differential diagnosis is wide, the histological findings may be subtle and relatively non-specific in early stages, and clinical findings may be extremely heterogeneous. Although MF is a low-grade lymphoma, it can involve lymphnodes and organs in about 10% of cases.

## 2. The Staging of Mycosis Fungoides

In 2007, the International Society for Cutaneous Lymphomas (ISCL) and the EORTC proposed a revision of the staging criteria, which were subsequently validated by a single-center study cohort conducted on 1502 patients. The National Comprehensive Cancer Network (NCCN) adapted the revised ISCL/EORTC recommendations for staging MF (Table 1 and Table 2).

**Table 1 healthcare-11-00614-t001:** International Society for Cutaneous Lymphomas/European Organization for Research and Treatment of Cancer classification of mycosis fungoides/Sézary syndrome.

TNMB Stages Definition	
Skin (T)	
T1	Patches, papules, and or plaques covering <10% body surface area (BSA) T1a patch only T1b plaque ± patch
T2	Patches, papules, and/or plaques covering ˃10% BSA T2a patch only T2b plaque ± patch
T3	One or more tumors (at least one 1 cm diameter) solid or nodular lesion with evidence of depth and/or vertical growth
T4	Confluence of erythema covering ˃80% BSA
NODE (N)	
N0	No clinically abnormal peripheral lymph nodes, biopsy not required
N1	Clinically abnormal peripheral lymph nodes, histopathology Dutch grade 1 or NCI LN_0–2_ (Table 2) N1a clone negative N1b clone positive
N2	Clinically abnormal peripheral lymph nodes, histopathology Dutch grade 2 or NCI LN_3_ (Table 2) N2a clone negative N2b clone positive
N3	Clinically abnormal peripheral lymph nodes, histopathology Dutch grade 3–4 (Table 2) or NCI LN_4_ (Table 3), clone positive or negative
Nx	Clinically abnormal peripheral lymph nodes, no histopathologic confirmation
VICERAL (M)	
M0	No visceral organ involvement
M1	Visceral involvement (must have pathology, and organ is to be specified)
BLOOD (B)	
B0	Absence of significant blood involvement: <5% of blood lymphocytes are Sézary cells B0a clone negative B0b clone positive
B1	Low blood tumor burden: >5% of peripheral blood lymphocytes are Sézary cells but does not meet criteria for B2 disease B1a clone negative B1b clone positive
B2	High blood tumor burden: ˃1000/µL Sézary cells with positive clone in blood (matching clone in skin) or positive clone.
DUTCH GRADE SYSTEM	
Grade1	Dermatopathic lymphadenopathy (DL)
Grade2	Early involvement by mycosis fungoides (MF), presence of cerebriform nuclei larger than 7.5 μm
Grade3	Partial effacement of lymph node architecture; many atypical cerebriform mononuclear cells (CMCs)
Grade4	Complete effacement of lymph node architecture
NCI LN GRADE SYSTEM	
LN0	No atypical lymphocytes
LN1	Occasional and isolated atypical lymphocytes not arranged in clusters
LN2	Many atypical lymphocytes or in 3–6 cell clusters
LN3	Many atypical lymphocytes or in 3–6 cell clusters
LN4	Partial/complete effacement of nodal architecture by atypical lymphocytes or frankly neoplastic cells

**Table 2 healthcare-11-00614-t002:** WHO/EORTC staging of MF/SS. Clinical stages and 5-year disease-free survival (DSS).

STAGE	T	N	M	B	5-Year DSS (%)
IA	1	0	0	0,1	98
IB	2	0	0	0,1	89
IIA	1,2	1,2	0	0,1	89
IIB	3	0–2	0	0,1	56
IIIA	4	0–2	0	0	54
IIIB	4	0–2	0	1	48
IVA1	1–4	0–2	0	2	41
IVA2	1–4	3	0	0–2	23
IVB	1–4	0–3	1	0–2	18

**Table 3 healthcare-11-00614-t003:** Treatment, according to the stage of the disease.

Stage I Stage II	➢ UVB-Narrow Band (mainly T1a and T2a) ➢ PUVA (thicker plaques) ➢ Topical corticosteroids ➢ ECP ➢ Topical chemotherapy: mechlorethamine, gel (Valchlor), gel (Ledaga^®^) ➢ Systemic chemotherapy with one or more drugs: retinoids, INF-alpha, low-dose methotrexate, TSEB ➢ Other drug therapy (lenalidomide, histone deacetylase inhibitors) ➢ Targeted therapy: brentuximab vedotin, mogamulizumab
Stage III Stage IV	➢ UVB-Narrow Band ➢ PUVA (thicker plaques) ➢ Topical chemotherapy: mechlorethamine, gel (Valchlor), gel (Ledaga^®^) ➢ ECP ± RI total skin electron beam radiation therapy ➢ RT total skin electron beam radiation therapy. ➢ Systemic chemotherapy with one or more drugs: retinoids, INF-alpha, low-dose methotrexate, TSEB ➢ Other drug therapy (lenalidomide, bexarotene, histone deacetylase inhibitors, pembrolizumab) ➢ Targeted therapy: brentuximab vedotin, mogamulizumab ➢ High-dose chemotherapy, and sometimes RT, with stem cell transplant.

Psoralen and ultraviolet A (PUVA), extracorporeal photopheresis (ECP), radiation therapy (RT), inteferona (INF), total-skin electron beam therapy (TSEBT).

The clinical stage is an important factor in determining the risk of disease progression (RDP) and overall survival (OS).

Skin directed therapy with the addition of systemic drugs are necessary in case of advanced disease, including the presence of tumors, erythroderma, and nodal, visceral, or blood involvement.

Patients affected by MF have long pathological stories with possible phases of remission and progression, and they should be at the center of a medical team with different competences. As MF is a challenge in terms of both diagnosis and therapy, the multidisciplinary team approach (MDTA) is mandatory to ensure a correct diagnosis and the best treatment choices.

## 3. Clinical Features

MF is a heterogenous disease, and cutaneous lesions are often polymorphic, making the clinical diagnosis particularly difficult. MF can mimic a wide range of dermatological diseases and differential diagnosis includes more than 50 different clinical entities [2,3].

In detail, differentiation from inflammatory skin diseases such as eczema and psoriasis or other dermatoses can be challenging and not rarely the persistent nature of the disease is a very important diagnostic element.

Histology in not always the decisive factor in early phase and the dermatologist plays a key role in guiding diagnostic decisions.

The clinical presentation of classical MF typically occurs in sun-protected areas (“bathing suit” distribution) with three stages: patch, plaque, and tumor, characterized by variability in the size, shape, and color of lesions.

Most patients with MF present a prolonged, indolent clinical course with initial skin involvement.

The disease starts with patches, which after years or even decades can develop into thin and thick plaques or tumors, but many patients initially present with thick plaques and even tumors (Figure 1).

The dissemination to lymph nodes, blood, bone marrow, and organs can be found. Relatively specific for early MF is the presence of poikiloderma, clinically defined as the local juxtaposition of mottled pigmentation, telangiectasia, and epidermal atrophy (cigarette paper wrinkling) interspersed with slight infiltration, but it is rare. Between all affected patients, we can count that about 71.5% are in early stage, 28.5% are in advance stage, and 9.7–11.6% are in intermediate stage [4,5,6,7].

Clinical signs of erythrodermal MF are redness and peeling of the entire skin with itching and pain. This must be distinguished mainly from leukemic Sezary syndrome with erythroderma. Large cell transformation (LTC) is a histopathological phenomenon with largely unclear clinical implications, and in the advanced stages, it can develop in between 8% and 23% of the patients [8,9]. LTC is defined by the evidence of sheets of large and atypical lymphocytes which exceed 25% of the tissue lymphocyte infiltrate. LTC is associated with a more aggressive disease course and significantly lower median survival (3.6 years) compared to non-transformed MF (8.8 years) [10,11].

According to the WHO-EORTC (2018) classification of primary cutaneous lymphomas, we recognize three distinct subtypes of MF with different histological features, clinical behavior, and/or prognosis: folliculotropic MF (FMF), pagetoid reticulosis, and granulomatous slack skin [12]. FMF accounts for 10% of all MF, while pagetoid reticulosis and granulomatous slack skin are extremely rare [13]. FMF differs from the classic form of MF by the presence of folliculotropic lymphocytic infiltrates often sparing the epidermis.

FMF is clinically characterized by predilection for head and neck with eyebrow involvement and the presence of grouped follicular papules, acneiform lesions, and associated alopecia. The FMF appears less responsive to several first-line skin-directed therapies such as psoralen plus ultraviolet light A (PUVA) and has a more aggressive clinical course compared with classical MF. Therefore, it may require more aggressive treatments. To date, two distinct subgroups of this disease have been identified, with different prognostic implication. Depending on whether the FMF is early stage or tumor/advanced stage, we can estimate five-year survival rates of 94% and 69%, respectively [14,15,16]. Recognition of indolent and more aggressive subgroups of FMF is mandatory, as it guides therapeutic choice and follow-up of the patient.

Pagetoid reticulosis is a rare variant of MF characterized by localized patches or plaque with an intraepidermal proliferation of atypical pagetoid lymphocytes singly or arranged in nests [17]. Pagetoid reticulosis includes a localized type and a disseminated type. The former (localized or “Woringer-Kolopp” type) presents with a solitary psoriasiform or hyperkeratotic plaque usually located on the extremities. This form shows an indolent course and has usually a good prognosis [18]. The preferred treatment is radiotherapy or surgical excision, but topical steroids and nitrogen mustard are acceptable therapeutic alternatives. The disseminated type (“Ketron-Goodman” type) is generally considered an aggressive lymphoma with poor prognosis like the tumor stage of MF and presents disseminated lesions with high rate of recurrence [19].

Granulomatous slack skin (GSS) is an extremely rare variant of MF with indolent course characterized by circumscribed areas of pendulous lax skin in the major skin folds with a predilection for the axillae and inguinal regions. Histologically, it shows a dense granulomatous dermal infiltrate containing atypical clonal T cells. In over 50% of cases, there is an association with Hodgkin’s lymphoma [20].

In recent years, dermoscopy has identified several patterns of MF that can aid the clinician in differential diagnoses with other inflammatory dermatoses. In the early patch stage, MF lesions show fine short linear vessels, orange-yellow patchy areas, and spermatozoa-like vascular structures. In FMF, we found folliculocentric erosions surrounded by dotted and fine linear vessels, comedo-like openings and perifollicular accentuation, and loss of terminal follicles [21]. Although dermoscopy is not relevant for diagnosis of MF, it can sometimes facilitate the diagnostic suspicion of MF and its differentiation from other inflammatory dermatoses.

Leukemic involvement is a rare but possible finding in MF, which needs to be differentiated from Sézary syndrome. The latter is a rare variant of cutaneous T-cell lymphoma characterized by erythroderma, leukemic involvement, and an absolute Sézary cell count of 1000 cells/mm^3^ or more.

The multiple variants of MF and its polymorphous clinical aspects necessarily require expert dermatological management with a permanent collaboration of dedicated dermatopathologists.

## 4. The Role of Histology in MF Diagnosis: Clues and Boundaries

Histological evaluation is always mandatory when MF is clinically suspected. Histology plays two main roles: to establish the diagnosis and to define the phase and the subtype of the neoplasm. Classic MF evolve in patches, plaques, and tumors, and the histological findings may vary according to the phase. Moreover, several variants of MF are defined on a histological point of view, including FMF, granulomatous MF and granulomatous slack skin syndrome, bullous MF, interstitial MF, poikilodermatous MF, syringotropic MF, anetodermic MF, and large cells transformation of MF [22,23].

The list of subtypes provided contains not only histopathological variants but rather subtypes that can only be diagnosed by dermato-pathological correlation. Although the morphology of MF is widely variable depending on the stage, the histological clues for histological diagnosis of MF generally include epidermal changes, epidermotropism of lymphocytes in single elements and/or aggregates (Pautrier’s microabscesses) without consensual spongiosis, ‘‘haloed’’ intra-epidermal lymphocytes, alignment of lymphocytes along the dermal-epidermal junction, larger dimension of intra-epidermal lymphocytes then intra-dermal lymphocytes, and hyperchromatic and irregular nuclei of lymphocytes (Figure 2).

Epidermal may show a variable appearance, ranging from hyperplastic to atrophic, and parakeratosis is frequent [24]. Attention must be paid in differentiating Pautrier’s microabscesses from intraepidermal inflammatory cells aggregated often present in chronic eczematous dermatitis. The former is constituted only by lymphocytes, with or without evident nuclear irregularities, without spongiosis in the adjacent epidermis. The latter is constituted by both lymphocytes and dendritic cells, with evident spongiosis in the adjacent epidermis. In tumor stage, epidermotropism is less evident or totally absent, and a diffuse nodular infiltrate of atypical lymphocytes throughout the dermis is the most frequent finding [25].

Establishing the diagnosis of MF in the early stage is certainly the greatest boundary in histological evaluation of MF, because the differential diagnosis is wide, including several different entities both reactive and malignant, and because the criteria for the histological diagnosis of MF are not entirely applicable in the earliest phases of the disease. The histological features of early MF are indeed subtle and generally non-specific in a high percentage of cases. Consequently, multiple biopsies performed at different moments of the disease course are often necessary. In 2015, Skov et al. retrospectively studied the history of 157 patients affected by MF to investigate the role of multiple biopsies and the time needed for the histological diagnosis [26]. The authors reported that a definite histological diagnosis of MF was made in only 25% of cases at the first biopsy, and three biopsies were necessary in 22% of cases. Interestingly, a histological diagnosis of MF was never established in 29% of cases, despite the fact that more than one biopsy was performed. In their classical study, Massone et al. evaluated the histological findings of early MF in a large series of 745 skin biopsies [27]. The most frequently observed histological findings resulted non-specific, including normal epidermis (48% of cases), focal interface dermatitis (59%), patchy-lichenoid dermal infiltrate (66%), and fibrosis/coarse collagen bundles in the papillary dermis (97%). The most characteristic morphological findings showed instead a low sensitivity, as single lymphocytes epidermotropism, basilar lymphocytes alignment, and Pautrier’s microabscesses were observed only in 22%, 23%, and 19% of the cases, respectively. Furthermore, cellular atypia was present only in about 10% of cases. The presence of “haloed” intra-epidermal lymphocytes is observed in 40% of the cases.

On the other hand, in 2000, the European Organisation for Research and Treatment of Cancer’s Cutaneous Lymphoma Study Group addressed the issue of histologic definition of early MF [28]. The panel of experts identified cytological atypia, defined as presence of medium-sized to large and cerebriform mononuclear cells, as the most important diagnostic finding, with a sensitivity and specificity between 90% and 100%. Other important diagnostic findings included linearly arranged single lymphoid cells closely related to basal keratinocytes, absence of dermal blast-like cells, and absence of fibrosis in the papillary dermis. However, the diagnostic value of lymphoid atypia is debated and showed discordant results in the different studies. Some attempts have been made in the last two decades to define shared and objective criteria for the diagnosis of early MF. In 2001, Guitart et al. proposed an integrated grading system including major criteria (density of the infiltrate, nature, and extent of epidermotropism, and grade of lymphocytic atypia) and minor criteria (presence of atypia primarily in the intraepidermal compartment, lack of associated inflammatory features, and reticular/wiry fibroplasia within the papillary dermis) [29]. The authors proposed four diagnostic categories based on the score obtained by the evaluation of these criteria: perivascular/interface lymphocytic dermatitis; atypical lymphocytic infiltrate (MF cannot be excluded); atypical lymphocytic infiltrate suggestive of MF; and MF (total score 7).

In 2005, Pimpinelli et al. proposed a scoring system for the diagnosis of early MF, taking into consideration clinical, histological, immunohistochemical, and molecular findings. Histological findings included the presence of a superficial lymphoid infiltrate (which was defined as basic criteria) and epidermotropism without spongiosis and lymphoid atypia, defined as enlarged and hyperchromatic nuclei and irregular or cerebriform nuclear contours (which were defined as additional criteria). Immunohistochemistry plays an ancillary role in diagnosis of early MF, as one or more pan-T antigen may be lost by the neoplastic lymphocytes. In this setting, the most useful findings include loss of CD2, CD3, or CD5 in more than 50% of the cells; loss of CD7 in more than 90% of the cells; and discordance in expression of one or more pan-T marker in epidermal lymphocytes compared to the dermal lymphoid population. These histological findings have good specificity for the diagnosis of MF, but are infrequent and consequently have poor sensibility, comprised between 10% and 60% [30]. Other immunohistochemical markers are being studied, but their application in MF diagnosis needs further confirmation. Recently, the immunohistochemical expression of vascular endothelial growth factor C (VEGF-C) appeared significantly higher in MF than in chronic benign dermatoses [31]. FK506-binding protein 51 (FKBP51), an immunophilin initially cloned in lymphocytes, resulted significantly expressed in MF in a recent study on 44 patients [32]. The application of molecular biology for the assessment of T-cell receptor (TCR) gene clonal rearrangements has recently improved the diagnostic efficacy of histology in MF. Detection of TCR gamma and beta gene rearrangement in histological samples suspected of MF can be of value, as neoplastic lymphoid proliferations are virtually always constituted by a clonal population. Nevertheless, molecular biology is still poorly applied worldwide in the diagnosis of MF, also because the needed technology is not still widely available in all surgical pathology laboratories. Comfere et al. conducted a survey on 144 expert dermatopathologists and reported that only 2.3% of them order TCR clonality assays as part of the initial diagnostic evaluation in cases of MF [32,33].

Molecular biology has a high impact on the diagnosis. The sensitivity of molecular tests is indeed not entirely satisfactory and is quite variable, ranging from 50% to 90% in various studies [34,35]. This may be due to both disease-correlated features (paucity of neoplastic cells in early stages) and technique-correlated features (method of assay, primer design, PCR products) [36]. On the other hand, TCR clonality is not entirely specific of lymphoma, as a clonal lymphocytic population may be present also in reactive dermatitis [37,38]. In this setting, establishing the same TCR clone in different biopsies of the same patients may be useful to distinguish MF from dermatitis [39]. The histopathological difficulties in the diagnosis of MF, therefore, require a solid collaboration between the pathologist and the clinical dermatologist, who is a key figure in the diagnostic management of the patient with cutaneous lymphoma.

### 4.1. Pathogenesis: New Insights and Molecular Markers

MF is a neoplastic proliferation of T cells homing in the skin, and its exact etiology is largely unknown. However, there have been several advances in the knowledge of the molecular biology of this neoplasm in the last years. In detail, some cytogenetic abnormalities have been reported, including loss of chromosomes 1p, 17p, 10q, 10, 13q, and 19 [40]. Genetic mutations lead to a dysregulated cell cycle control mainly involving the regulation of apoptosis. Decreased and/or defective expression and activity of Fas, a receptor belonging to the tumor necrosis factor family, seems to play a crucial role in this setting [41,42,43,44]. The neoplastic cells also aberrantly express cFLIP, which contributes to impaired Fas-mediated apoptosis [45]. Overall, genetic alterations in different molecules involved in the cell cycle regulation have been described in MF, such as the amplification of JUNB [46,47,48], constitutive activation of STAT3 [49,50], and decreased expression of p14, p15, and p16 [51,52]. Recent evidence highlights the role of the skin microenvironment in the development of MF. It is assumed that the neoplastic transformation of the lymphocytes in MF arises in the background of chronic inflammation. In particular, mast cells seem to play a protumorigenic role in cutaneous lymphomas [53], while reactive CD8-positive cytotoxic T cells, dendritic cells, and macrophages may contribute to an antitumor response [54,55,56].

### 4.2. Staging

The diagnosis of MF needs the comprehensive evaluation of clinical, histological, and immunophenotypical findings of every single case, performed by specialists with different expertise. However, the MDA is essential not only for the diagnosis but also for the correct management of the patient and treatment of the disease. All patients, for prognostic and therapeutic purposes, should be staged to assess the type of lesions and the percentage of body surface area involved, and to exclude the presence of extracutaneous localization in lymph nodes, viscera, and blood, based on TNMB staging system (Table 1).

The clinical stage is an important factor in determining RDP and OS (Table 2).

Physical examination and skin biopsy should be followed by blood cell count, routine blood chemistry tests with lactate dehydrogenase (LDH), and appropriate imaging studies (computed tomography (CT) and positron emission tomography (PET)) [57]. In case of leukemic forms, the peripheral blood smear with microscopic blood examination can demonstrate large lymphocytes with grooved, lobulated, “cerebriform” nuclei (Sezary cells). Immunophenotypically, Sezary cells show positivity for CD3 and CD4 and negativity for CD8 and CD7. The aberrant loss of one or more T-cell markers (CD2, CD3, CD4, CD5, and CD7) and the aberrant expression of MHC class I binding, killer immunoglobulin-like receptor (KIR) CD158k (normally expressed by natural killer (NK) cells) are frequently observed. The loss of CD7 and an overexpression of CD26 is sensitive and specific. Bone marrow examination is usually not mandatory in patients with MF [58].

In stage IA of the disease, no laboratory or imaging studies are required. Independent unfavorable prognostic factors in MF include older age, large cell transformation, and increased LDH values [59,60,61]. In case of early MF, the patient is usually managed by the dermatologists [62].

### 4.3. Treatment

Treatment goals in advanced MF (stage > IIB) include reducing the burden of disease (including management of pruritus), delaying progression, and improving or preserving quality of life, under the care of a hematologist/oncologist with the solid collaboration of the dermatologist. According to the NCCN recommendations for the treatment of MF, in patients with stage IA (only patches and/or plaques covering <10% of the skin surface), IB (only patches and/or plaques covering >10% of the skin surface), and IIA MF, there is a choice between watchful waiting (mainly T1a) and the use of skin directed therapies such as topical corticosteroids (mainly T1a and T2a) (Table 3).

UVB-Narrow Band (mainly T1a and T2a) is recommended for patients with patches or very thin plaques, but PUVA is preferred for patients with thicker plaques [63,64]. Mechlorethamine is a chemotherapeutic alkylating agent approved in multiple countries for topical dermatologic treatment of MF in adult patients. It prevents DNA synthesis and RNA transcription by attaching alkyl groups to DNA bases, forms cross-links on DNA, and mispairs nucleotides which causes mutations. Multiple studies show that topical mechlorethamine is an effective and safe therapy for MF. The efficacy of MCH is around 51–84% CR for patients with stage T1 MF and 31–62.2% for T2 MF disease [65] (Table 3).

The US Food and Drug Administration approved MCH 0.016% gel (Valchlor) for stage IA and IB MF in 2013 for patients who had received other SDTs first. In 2017, the European Medicines Agency (EMA) approved MCH 0.02% gel (Ledaga^®^) for treatment in MF [66]. De Quatrebarbes et al. showed that twice-weekly applications with MCH 0.02% aqueous solution followed by a potent TCS for 6 months in early-stage MF achieved a 58% CR rate [67]. The gel is applied once a day to affected areas of the skin and shows frequent adverse reaction: skin irritation, erythema, rash, urticaria, skin-burning sensation, cutaneous pain, pruritus, skin infection, skin ulceration, blistering, and hyperpigmentation [68]. No increased risk of NMSC or melanoma skin cancers was found [69] (Table 3).

Systemic therapies, most combined with PUVA, are second-line treatment of stages IA, IB, and IIA MF and include retinoids, interferon (INF)-alpha, total-skin electron beam therapy (TSEBT) (mainly T2b), and low-dose methotrexate (MTX) p.o. or s.c, only rarely 10–20 mg once weekly i.m.

In patients with stage IIB MF, retinoids and INF-alpha are recommended and are frequently combined with PUVA. (Table 3).

Other first-line treatments of stage IIB are gemcitabine and pegylated liposomial doxorubicin in monotherapy, low dose MTX, and localized radiotherapy (RT). 

The second line treatment of stage IIB MF is polychemotherapy CHOP (cyclophosphamide, doxorubicin, vincristine, prednisone), which is the most widely used regimen. However, several variants and other combinations are available: CEOP (cyclophosphamide, epirubicin, vincristine, prednisone), CVP (cyclophosphamide, epirubicin, vincristine, prednisone), and COMP (cyclophosphamide, liposomial doxorubicine, vincristine, prednisone) [70,71,72]. Allogenic stem cell transplantation should be restricted to selected patients.

The first line treatment in the stage IIIA and B MF is systemic therapies with retinoids and INF-alpha combined with PUVA, ECP used alone or in combination with skin-directed and other systemic drugs (retinoids and INF-alpha), low dose MTX, and TSEBT (Table 3).

The second line treatment is monochemotherapy with gemcitabine or pegylated liposomal doxorubicine. The liposomal pegylated formulation of doxorubicin remains in circulation for a long time. Pegylated liposomes contain surface bonded segments of the hydrophilic polymer methoxypolyethylene glycol (MPEG). These linear groups extend from the liposome surface creating a protective coating that reduces interactions between the two-layer lipid membrane and plasma components. This allows the liposomes to circulate for extended periods in the blood. Pegylated liposomes are small enough (average diameter of about 100 nm) to pass intact (by extravasation) through the fenestrated capillaries that supply the tumors. Standard doxorubicin hydrochloride exhibits significant tissue distribution (volume of distribution 700 to 1100 l/m^2^) and rapid elimination clearance (24 to 73 l/h/m^2^). For equivalent doses, the plasma concentration of liposomal doxorubicin is higher than the values obtained with standard preparations of doxorubicin hydrochloride (Table 3).

Allogeneic stem cell transplantation should be restricted to exceptional patients. Recommendations for treatment of MF stages IVA and IVB are gemcitabine, pegylated liposomal doxorubicine, CHOP and CHOP-like polychemotherapy, radiotherapy, TSEB and localized (used alone or in combination with systemic therapies), alemtuzumab (mainly in B2), and allogenic stem cell transplantation. In patients requiring maintenance therapy after disease remission, the use of topical corticosteroids, mechlorethamine, PUVA, UVB-NB, retinoids, INF-alpha, low dose of MTX, and ECP can be evaluated (Table 3).

Targeted immunotherapy represents an increasingly present reality for MF treatment, with increasingly satisfactory results. Among the most used drugs, we have mogamulizumab (anti-CCR4 CC chemokine Receptor 4), denileukindiftitox (anti-CD25/IL-2 receptor), alemtuzumab (anti-CD52), brentuximab vedotin (anti-CD30), histone-deacetylase inhibitors (HDAC inhibitors: vorinostat; panobinostat, romidepsin; belinostat, resminostat), immune checkpoint inhibitors (nivolumab; pembrolizumab), IPH4102 (anti-KIR3DL2 antibodies), lenalidomide, everolimus (oral mTOR inhibitor), and duvelisib (oral PI3K inhibitor) [73,74,75,76]. Among these, many are still experimental and not widely used. Mogalizumab is a humanized IgG1κ monoclonal antibody with defucosylated Fc region, which binds selectively to CCR4. The antibody carries out its oncolytic activity through antibody-mediated cytotoxicity on target cells. The drug was shown to be more effective in SS rather than MF; the ORR was 21% in MF and 37% in SS in a phase 3 randomized international study (MAVORIC Study), compared to Vorinostat which instead showed an ORR of 4% [77,78,79]. In all studies, the drug showed an acceptable safety profile, with common toxicities including nausea, headache, fever, diarrhea, itching, and infusion reactions. This drug is new in the field and thus not widely used. (Table 4)

Denileukindiftitox CD25-directed cytotoxin indicated for the treatment of patients with persistent or recurrent cutaneous T-cell lymphoma whose malignant cells express the CD25 component of the IL-2 receptor has been unavailable for many years. Alemtuzumab is a genetically engineered humanised IgG1 kappa monoclonal antibody specific for a 21–28 kD cell surface glycoprotein (CD52) of the lymphocyte.

Brentuximab Vedotin is a drug-conjugated antibody composed of an anti-CD30 monoclonal antibody; according to some recent phase II-III studies, it has reached an ORR of 70% in patients with MF/SS CD30+, and it is now commonly used starting from the second line of systemic therapy.

Vorinostat, Romidepsin, and Belinostat act by blocking the activity of histone deacetylases involved in gene activation and deactivation and have been shown in studies conducted in refractory/relapsed cutaneous T-cell lymphomas after the second lines of systemic ORR therapy about 30% [79,80] (Table 4).

Among the recently used drugs, we can mention pralatrexate (folate analogue) which has a greater activity than methotrexate thanks to its greater affinity for reduced folate carrier 1, which allows a greater accumulation in cancer cells, and forodesine which is a potent inhibitor of purine nucleoside phosphorylases (PNP) that causes apoptosis in neoplastic T cells and normal T lymphocytes, sparing other lymphocyte populations [81] (Table 4).

Bortezomib, a proteasome inhibitor commonly used in myeloma, also showed an average ORR of 67% and CR of 17% in MF. Although there are several therapies identified by the NCCN for the treatment of MF, only a few treatments can give lasting answers. Targeted therapies have a wide variability of responses ranging from 30% to 67%, with complete responses that do not exceed 41%. Conventional chemotherapy, although it can achieve a high response rate, has short-lived results, and is associated with worse overall outcomes. The only potentially eradicating treatment for MF is non-myeloablative allogeneic transplantation, which, however, has only 46% OS at 5 years (Table 4).

Recently, the Stanford transplant regimen showed an overall response rate of 90%, with a two-year OS of 76% and a two-year OS of 50%. Given the limited efficacy of available therapies, advanced patients should be encouraged to participate in clinical trials. 

Several agents are currently being studied for the treatment of cutaneous lymphomas including MF. These include E7777 (cytotoxic IL-2 fusion protein), MRG-106 (miR-155 antagonist), Duvelisib (PI3K inhibitor), Ruxolitinib (JAK 1/2 inhibitor), and TTI-621 (SIRPaFc IgG4, anti-CD47).

The approach to relapsed/ refractory patient to the first line of chemotherapy is very complex. The patient with MF is more sensitive to infections but after several therapies more and more. It is a tough challenge because most of the MF patients we lose die from infections. The new therapies described, and the target therapy allow us to treat these patients with hope of a response. In particular, target therapy often allows us to treat patients with several comorbidities and a poorer performance status, with greater guarantees in avoiding neutropenias which can be particularly bad. This is a big advantage over the past.

The role of the radiotherapist in the treatment of MF has historical roots dating back to 1902–1903 [88,89,90,91]. The scientific rationale for use of radiotherapy (RT) is due to the high radiosensitivity of MF cells that often regress even with low-dose RT, minimizing the side effects of the therapy. The correlation between dose and DFS increase has been documented by Kim et al. in 1976 [92]. The increase in dose, however, also leads to an increase in side effects that also depend on the irradiated anatomical site. The most common side effects reported are erythema, pain at the irradiated site, and alterations of the blood count. RT can be used for the treatment of MF with curative/primary, adjuvant, and palliative intent. It is advisable, given the rarely occurring disease, the MDA, involving hematologists/oncologists, dermatologists, and radiation oncologists. The general intent of radiant treatment is to clear visible skin disease with an adequate margin that considers microscopic disease and deep localizations. For localized lesions, involved site RT (ISRT) is used. For RT treatment, planning a TC simulation scan is required, using immobilization systems comfortable for the patient and eventually a thermoplastic bolus for dose superficialization. In the contouring phase, target volumes and surrounding organs at risk are evaluated using available diagnostic imaging. The gross tumor volume (GTV) is represented by the visible disease, to which a margin is added to define the clinical target volume (CTV) that encompasses the microscopic disease. A further expansion is added to the CTV to define the planning target volume (PTV) to consider variation in patient position, organ motion, and other uncertainties. During the planning phase, the radiation to be used (usually electrons for superficial lesions or X-rays for deeper lesions) is decided. The conformational 3D technique (3D-CRT) remains the standard, reserving special techniques such as intensity modulated RT (IMRT) for particularly complex localizations such as the orbital brain region [93]. RT with conventional fractionation (1.8–2 Gy) with doses ranging from 24 to 30 Gy, usually with 6–9 MeV electrons, 5 days a week, is recommended. Higher doses may be reserved for particularly large or deep disease localization. For the treatment of larger areas, TSEBT can be safely used. TSEBT is a complex technique that should be reserved to high expertise centers. Patients are generally treated standing on dedicated platforms to increase the coverage of the areas to be irradiated. The conventional fractionated RT dose range is 12–36 Gy, generally 4–6 Gy per week [86,87,94]. Adjuvant RT can be used after the failure of local or systemic therapies or in case of relapses with doses to be chosen according to the clinical presentation and/or clinical condition of the patient. Palliative radiotherapy can play a cytoreductive and/or antalgic role, with fractionations usually used in other palliative settings.

In Italy and in many other European countries, the role of the transfusionist in the management of MF is mainly related to extracorporeal photochemotherapy (ECP) [95]. ECP is an immunomodulatory therapy, which is characterized by the extracorporeal exposure of mononuclear cells (MNC), collected from the patient’s peripheral blood through a cell separator, to the irradiation of ultraviolet light (UVA) in the presence of a photosensitizing agent 8-methoxypsoralene (8-MOP) [96]. 8-MOP is a biologically inert derivative of furocoumarin which enters cell nuclei and remains biologically inert until exposure to UVA 2. After exposure, it becomes able to establish covalent and cross-linked links with the DNA double helix and to bring cell apoptosis [97]. The ECP induces apoptosis of the lymphocytes treated within 24–72 h after the treatment. Furthermore, ECP induces involvement of antigen-specific T-regulatory cells (T-regs) from “native” T cells through the secretion of anti-inflammatory cytokines such as IL-10 and TGF-b [98,99,100]. Hence, ECP induces an immune-mediated response against the malignant T-cell clone, not only inducing apoptosis but promoting the presentation of tumor-loaded dendritic cells (DCs) to cytotoxic T cells, the conversion of monocytes into immature DCs, and the expansion of a cytotoxic T cell population against the malignant T cell clone [101,102,103,104].

ECP is a treatment that involves three phases:Collection of leukocytes (leukapheresis) on the patient with enrichment of mononuclear cells [103];The addition of 8-MOP followed by exposure of the cell concentrate collected to irradiation with UVA rays (2J/cm^2^) [105];Reinfusion of the treated cells.

There are two systems to perform the ECP: “in-line” method, in which the three phases of the ECP (MNC collection, UV-A irradiation, and reinfusion to the patient) take place in the same equipment, and “off-line” method, in which the three phases are distinct and carried out by two different devices. The most common side effects are sporadic and mild, such as nausea, fever, or headache. The consistent feature across the available evidence is that ECP is known to be a distinctly safe form of therapy. Serious side effects, such as sepsis, occur rarely, which is extremely valuable in patients whose alternative treatments are highly toxic such as chemotherapy. In MF, clonal (malignant) epidermotropic CD3+/CD4+ T cells are involved [106,107,108,109,110].

The guidelines of the National Comprehensive Cancer Network (NCCN) give the broadest indications:ECP is recommended as second line for the treatment of MF stage IA, refractory to skin-directed therapies, alone or in combination with skin-directed therapies;As second line for the treatment of MF stage IB, IIA, and III refractory to skin-directed therapies;As first-line for the treatment of MF stage IIB alone or in combination with skin-directed therapies;As first-line for the treatment of SS stage IV.

The treatment should be started with one cycle (two consecutive days) every 2 weeks in the first 3 months, and then for the next 3 months, 1 cycle monthly [82,83,84,85,111,112]. Schedule should be continued for a period of not less than 6 months. In patients with good response or disease stabilization, treatment should not be stopped and should be continued for approximately more than 2 years, with intervals of up to 6–12 weeks and subsequent interruption if no relapse occurs [82,83,84]. The median time to maximal response on ECP is 5–6 months, although some patients may take up to 10 months to respond. In case of relapses, the ECP can be resumed monthly or fortnightly [82,112]. 

In all cases, progression of disease or absence of response should be considered to increase the frequency of treatments, combination therapy, or chemotherapy [82,83,84,85]. In these cases, ECP can usually be associated with IFN-α, IFN-γ, and/or systemic retinoid (bexarotene).

## 5. Conclusions

MF is characterized by a marked clinical polymorphism and a remarkable mimicry towards the main inflammatory dermatoses [102,113] and presents an indolent evolution which can last for decades. The diagnosis is consequently challenging, with significant delay for the start of therapy [113]. In addition, its progression with involvement of the entire body surface and systemic advancement makes it necessary in the management of the patients a multidisciplinary team of expert dermatologists, pathologists, and hematologists. Radiotherapists, radiologists, laboratory technicians, nutritionists, nurses, and psychologists are necessary to support the team members.

The multidisciplinary approach is the best way to deliver complex care in cancer patients. In the literature, this approach has been described since 1975, but it is only since 1997 that it entered in clinical practice with a continuous increase in use until it became routine for some types of cancer with the constitution and improvement of tumor boards, a multidisciplinary teams focused on neoplastic patient’s care. Many studies show that multidisciplinary care (MDC) can reduce mortality and improve quality of life in oncological patients and those therapeutic decisions made by multidisciplinary teams are more in accord with evidence-based guidelines than those taken by individual healthcare professionals. The combination of several specialists allows those who have more competence to intervene at the right time in a specific step of the therapeutic diagnostic path.

In addition, the multidisciplinary team (MDT) approach leads to a reduction in the number of outpatient visits and hospitalizations, improving economic performance and increasing both patient compliance and satisfaction. A disease as heterogeneous and complex as MF is the typical example to demonstrate how important MDC is in oncological pathologies.

## Figures and Tables

**Figure 1 healthcare-11-00614-f001:**
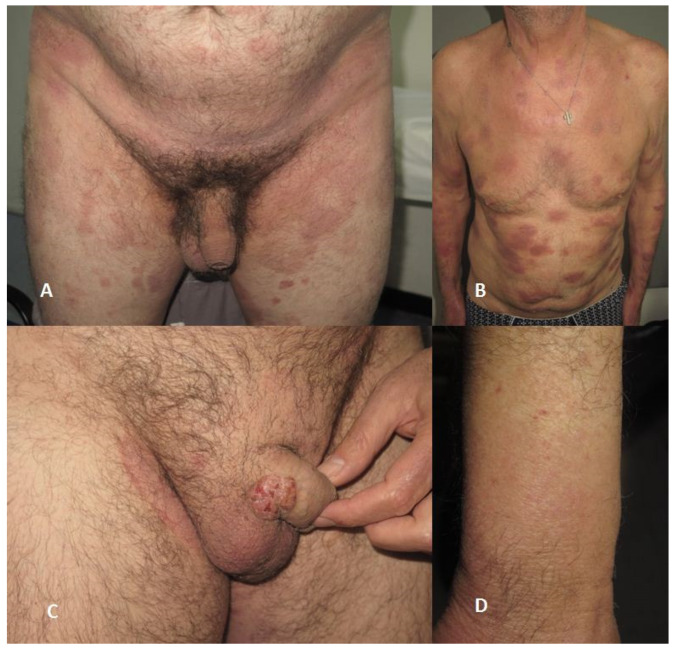
(**A**) Typical clinical presentation with erythematous patches in the bathing suit area. (**B**) Patient presenting multiple typical erythematous plaques on the trunk. (**C**) Genital tumor with superficial erosions. (**D**) Alopecic patch on forearm in patient with folliculotropic mycosis fungoides.

**Figure 2 healthcare-11-00614-f002:**
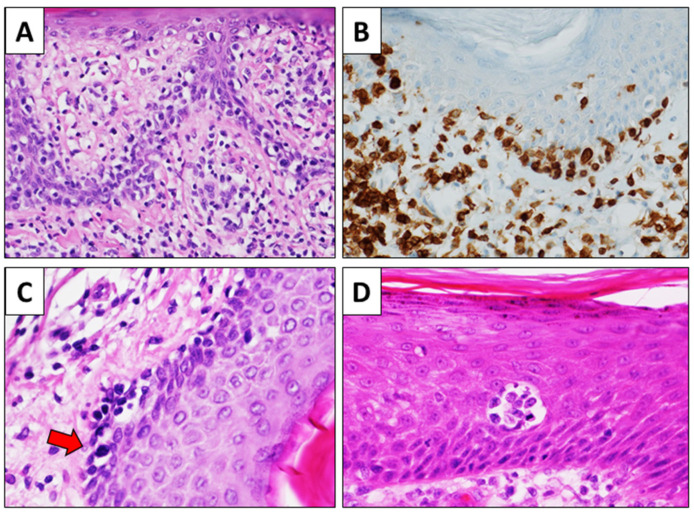
Histological clues of MF. Lymphocytes are aligned along the dermal-epidermal junction; they are surrounded by a halo and are larger than the dermal lymphocytes ((**A**), H&E, original magnification 100×). The alignment of lymphocytes is highlighted by CD3 immunohistochemistry ((**B**), CD3 immunostain, original magnification 200×). Some intraepidermal lymphocytes show irregular and hyperchromatic nuclei (red arrow) ((**C**), H&E immunostain, original magnification 400×). Pautrier’s microabscesses are characterized by aggregates of atypical lymphocytes in the thickening of the epidermis, which shows no significant spongiosis. In this image, a mitosis is also evident ((**D**), H&E, original magnification 200×).

**Table 4 healthcare-11-00614-t004:** Summary of recent clinical study.

Agent [Ref.]	Trial	CTCL Indication	Approval Year (FDA)	ORR%/CR%/PR%
Romidepsin [80,81]	Pivotal/supportive	Relapsed/refractory CTCL	2009	33.8/5.6/28.2
Denileukin diftitox [73,74,75,76]	Pivotal	Persistent/recurrent CTCL	1999	44/10/34
Bexarotene [82,83,84,85]	Pivotal	Refractory CTCL	1999	45
Vorinostat [80,81]	Pivotal/Supportive	Progressive/persistent/recurrent CTCL	2007	29.5
Brentuximab Vedotin [73,74,75,76]	Randomized trial vs methotrexate or bexarotene(ALCANZA)	pcALCL or CD30-expressing MF refractory	2017	56.3
Mogamolizumab [77,78,79]	Randomized trial vs vorinostat(MAVORIC)	Relapsed or refractory MF or SS	2018	28
Pembrolizumab [73,74,75,76]	Multicenter Phase II study	Advanced relapsed or refractory MF/SS	-	37.5/8.3/29.2
Cobomarsen(MRG-106) [77,78,79]	Phase II trialRandomized Trial(SOLAR) vs vorinostat	Progressive/recurrent/persistent MF	2020 orfan drug	
Alemtuzumab [73,74,75,76]	Phase II Study	Advanced/relapsed MF or SS	-	51.1/17.9/33.3
IPH4102 (anti-KIR3DL2 antibody) [73,74,75,76]	Phase I clinical trial	Relapsed/refractory MF/SS	-	
Pralatrexate [63,86,87]	Phase I clinical trial	Advanced/relapsed PTCL	2009	44.8/3.4/41.4
Forodesine [81]	Multicenter phase I–II study iv/os	Advanced/relapsed CTCL	-	ORR 31% iv 27% os
Duvelisib [73,74,75,76]	Open label phase I study	Relapsed or refractory CTCL	-	31.6/0/31.6
Lenalidomide [73,74,75,76]	Open label multicenter phase II study	Advanced/refractory MF/SS	-	PR 28%
Gemcitabine [63,86,87]	Multicenter phase II study	Advanced/refractory CTCL	-	75/21.8/53.1
Pegylated liposomal doxorubicin [63,86,87]	Multicenter study	Advanced relapsed/recurrent mf/ss	-	56/20/36

## Data Availability

Not applicable.

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
