# Peer review of "Multidisciplinary Approach to the Diagnosis and Therapy of Mycosis Fungoides"

_healthcare, 2023, doi:10.3390/healthcare11040614_

Round 1

Reviewer 1 Report

The Introduction contains a lot of details, making it too lengthy. Some information in the Information is also redundant to a number of statements in the Results and Discussion sections. Please make the Introduction shorter.

Author Response

To the Editor in Chief of Healthcare

We re-submit our article “Multidisciplinary Approach To The Diagnosis And Therapy Of Mycosis Fungoides”, Manuscript ID: healthcare-2172355.

The following changes (shown underlined). The manuscript has been improved according to the suggestions of the reviewer: I thank the Editor for the opportunity of reviewing this paper.

Reviewer(s)' Comments to Author:

Reviewer #1
: The Introduction contains a lot of details, making it too lengthy. Some information in the Information is also redundant to a number of statements in the Results and Discussion sections.

Point 1: Please make the Introduction shorter.

Answer to the Reviewer point 1: The observation of the reviewer has been accepted and the new manuscript has been  modified accordingly.

Reviewer #2: the manuscript is very interesiting and the multidisciplinary approach to the mycosis fungoides is very inspiring.

Point 1: Did the authours check out if any location in the CNS occur to analyse the invasiveness of the  pathology, that might lead to lymphoma? Thank you

Answer to the Reviewer point 1: The observation of the reviewer has been accepted. Mycosis Fungoides do not infiltrate the CNS. MF is predominantly localized to the skin and disease progression affects the lymphonodes, visceral organs, and blood. When infiltration involves blood with increased Sezary cells, MF turns into Sézary Syndrome. Sézary Syndrome can affect the CNS. Then differentiation is carried out at the diagnostic stage.

Reviewer #3: The review is well conducted and clear. It is uptodated and interesting, although the topic is a target of freguent reports.

Answer to the Reviewer point 1: We thank the Reviewer for reviewing our manuscript and for commenting.

Reviewer #4: The review is devoted to an important medical issue including diagnostic criteria, therapy options and prognosis of mycosis fungoides. Mycosis fungoides is one of the most common primary cutaneous T-cell lymphoma, characterized by skin-homing CD4+T cells derivation. There are multiple treatment regimes and various histological findings are described. The review will be useful for practitioners and residents in oncology and dermatology. Specific comments:

Point 1: There were sentences with unclear meaning ( lines 131-132, 241-243, 288-290.)

Answer to the Reviewer point 1: The observation of the reviewer has been accepted and the new manuscript has been  modified accordingly

Reviewer #5: Dear Authors, 

first of all I would to thank you for provided review with is really comprehensive. 

All my comments and suggestions I wrote as revisions inside the attached document. 

In brief: 

Point 1: . First of all I would advise to provide an English editing with a native speaker

Answer to the Reviewer point 1: The observation of the reviewer has been accepted and the new manuscript has been revised by English teacher.

Point 2:  I found that not everywhere you put references, especially in the tables with provided studies (see document attached)

Answer to the Reviewer point 2: The observation of the reviewer has been accepted and the new manuscript has been  modified accordingly.

Point 3: The part of staging and treatment: I would divide it. Moreover, it will be better to structure the therapy in this way - first-line treatment, relapsed/refractory options, novel approaches. 

Answer to the Reviewer point 3: The observation of the reviewer has been accepted and the new manuscript has been modified accordingly. The aim of the work is to underline the importance of multidisciplinary teams in the diagnostic, clinical, and therapeutic management of MF, highlighting the roles of individual specialists in the various stages of disease management and the importance of multidisciplinary intervention at other stages.

In the new manuscript we have better highlighted, for a more appealing reading of the reader, the first line treatment, the second line of treatment, relapsed / refractory options, novel approaches.

We thank the Editor and the Reviewers for helping us to improve our paper.

The manuscript has been read and approved by all the authors.

We also declare that we have no conflict of interest in connection with this paper.

We sincerely hope that the enclosed manuscript can be accepted for publication in the: Healthcare

Prof. C Sagnelli

Dr. A Sica

Reviewer 2 Report

the manuscript is very interesiting and the multidisciplinary approach to the mycosis fungoides is very inspiring. Did the authours check out if any location in the CNS occur to analyse the invasiveness of the  pathology, that might lead to lymphoma? Thank you

Author Response

(The authors gave the same response as above.)

Reviewer 3 Report

The review is well conducted and clear. It is uptodated and interesting, although the topic is a target of freguent reports.

Author Response

(The authors gave the same response as above.)

Reviewer 4 Report

 The review is devoted to an important medical issue including diagnostic criteria, therapy options and prognosis of mycosis fungoides. Mycosis fungoides is one of the most common primary cutaneous T-cell lymphoma, characterized by skin-homing CD4+T cells derivation. There are multiple treatment regimes and various histological findings are described. The review will be useful for practitioners and residents in oncology and dermatology. Specific comments: There were sentences with unclear meaning ( lines 131-132, 241-243, 288-290.)

Author Response

(The authors gave the same response as above.)

Reviewer 5 Report

Dear Authors, 

first of all I would to thank you for provided review with is really comprehensive. 

All my comments and suggestions I wrote as revisions inside the attached document. 

In brief: 

1. First of all I would advise to provide an English editing with a native speaker

2. I found that not everywhere you put references, especially in the tables with provided studies (see document attached)

3. The part of staging and treatment: I would divide it. Moreover, it will be better to structure the therapy in this way - first-line treatment, relapsed/refractory options, novel approaches. 

Author Response

(The authors gave the same response as above.)

Round 2

Reviewer 5 Report

Dear Authors, 

thank you for editing this your manuscript due my comments and suggestions. However, the part of treatment is still described in a really busy way. I will recommend to provide at least one table. On your choice. 

My best 

Author Response

To the Editor in Chief of Healthcare

We re-submit our article “Multidisciplinary Approach To The Diagnosis And Therapy Of Mycosis Fungoides, Manuscript ID: healthcare-2172355".

The following changes (shown underlined). The manuscript has been improved according to the suggestions of the reviewer: I thank the Editor for the opportunity of reviewing this paper.

Reviewer(s)' Comments to Author:

Reviewer #5
: Dear Authors,

thank you for editing this your manuscript due my comments and suggestions. However, the part of treatment is still described in a really busy way.

Point 1: . I will recommend to provide at least one table. On your choice

Answer to the Reviewer point 1: The observation of the reviewer has been accepted and the new manuscript has been  modified accordingly.

We thank the Editor and the Reviewers for helping us to improve our paper.

The manuscript has been read and approved by all the authors.

We also declare that we have no conflict of interest in connection with this paper.

We sincerely hope that the enclosed manuscript can be accepted for publication in the: Healthcare

Prof. C. Sagnelli
